# Adipose-Derived Stem/Stromal Cells in Kidney Transplantation: Status Quo and Future Perspectives

**DOI:** 10.3390/ijms222011188

**Published:** 2021-10-17

**Authors:** Gabriele Storti, Evaldo Favi, Francesca Albanesi, Bong-Sung Kim, Valerio Cervelli

**Affiliations:** 1Plastic and Reconstructive Surgery, Department of Surgical Sciences, Tor Vergata University, 00133 Rome, Italy; gabriele.stortimd@gmail.com (G.S.); valeriocervelli@virgilio.it (V.C.); 2Department of Clinical Sciences and Community Health, University of Milan, 20122 Milan, Italy; 3Kidney Transplantation, Fondazione IRCCS Ca’ Granda Ospedale Maggiore Policlinico, 20135 Milan, Italy; francesca.alba.96@gmail.com; 4Division of Plastic Surgery and Hand Surgery, University Hospital Zurich, 8091 Zurich, Switzerland; bong-sung.kim@usz.ch

**Keywords:** adipose stem cells, extra-cellular vesicles, kidney transplantation, ischemia–reperfusion injury, tolerance, rejection, acute kidney injury, regenerative medicine

## Abstract

Kidney transplantation (KT) is the gold standard treatment of end-stage renal disease. Despite progressive advances in organ preservation, surgical technique, intensive care, and immunosuppression, long-term allograft survival has not significantly improved. Among the many peri-operative complications that can jeopardize transplant outcomes, ischemia–reperfusion injury (IRI) deserves special consideration as it is associated with delayed graft function, acute rejection, and premature transplant loss. Over the years, several strategies have been proposed to mitigate the impact of IRI and favor tolerance, with rather disappointing results. There is mounting evidence that adipose stem/stromal cells (ASCs) possess specific characteristics that could help prevent, reduce, or reverse IRI. Immunomodulating and tolerogenic properties have also been suggested, thus leading to the development of ASC-based prophylactic and therapeutic strategies in pre-clinical and clinical models of renal IRI and allograft rejection. ASCs are copious, easy to harvest, and readily expandable in culture. Furthermore, ASCs can secrete extracellular vesicles (EV) which may act as powerful mediators of tissue repair and tolerance. In the present review, we discuss the current knowledge on the mechanisms of action and therapeutic opportunities offered by ASCs and ASC-derived EVs in the KT setting. Most relevant pre-clinical and clinical studies as well as actual limitations and future perspective are highlighted.

## 1. Introduction

Kidney transplantation (KT) is the gold standard treatment of end-stage renal disease (ESRD). In the last two decades, the overwhelming discrepancy between available organs and patients in the transplant waiting list has led to a progressive expansion of the criteria adopted for donors’ acceptance and allocation. As such, an increasing number of so-called marginal kidneys are now being transplanted into high-risk recipients. The main consequence of this aggressive policy is a striking rise in the incidence of delayed graft function (DGF) [1,2,3]. Clinically defined as the need for dialysis during the first post-transplant week, DGF is a complex entity that can negatively affect both patient and allograft survival [4]. The most relevant risk factors for DGF are expanded criteria (ECD) or donation after circulatory death (DCD) donors, prolonged warm (WIT) or cold ischemia time (CIT), and recipient sensitization [5]. Among the various peri-transplant events contributing to the development of DGF, ischemia–reperfusion injury (IRI) deserves special consideration as it is inevitable and represents the major determinant of acute tubular necrosis (ATN), the leading histological finding associated with DGF [2]. IRI is accompanied by a massive pro-inflammatory response and can trigger multiple cell death programs, endothelial dysfunction, transcriptional reprogramming, and activation of both innate and adaptive immunity [6]. Considering the strong association between IRI, DGF, acute rejection (AR), and progressive interstitial fibrosis with tubular atrophy (IF/TA) as well as their detrimental impact on allograft function and survival, the prevention and treatment of IRI have been recognized as primary issues for the transplant community [7].

Over the years, several strategies have been proposed to mitigate the effects of IRI and favor allograft acceptance, but to date, there is no specific therapy for DGF or tolerance induction approved by the Food and Drug Administration (FDA) or the European Medicines Agency (EMA) [6,8,9]. Depending on the target and timing of administration, possible interventions can be sorted into three main categories: donor-related, organ-related, and recipient-related.

Donor-targeted approaches include optimization of peri-operative care (hemodynamic stability and biochemical balance) and surgical technique during organ procurement, in order to reduce WIT, CIT, and direct damage to the kidneys [8,9]. Donor ischemic or hypothermic preconditioning [10,11] and specific pharmacological treatments such as anti-diuretic hormone (ADH) [12], catecholamines [13], corticosteroids [14], thyroid hormones [15], N-acetylcysteine [16], and erythropoietin [17] have also been tested in pre-clinical and clinical models of renal allograft IRI, showing mixed results.

Organ-related strategies focus on ex vivo kidney preservation and reconditioning. Static cold storage (SCS) represents the most widely used organ preservation modality. There are four main perfusion solutions designed for SCS: University of Wisconsin (UW), histidine-tryptophan-ketoglutarate (HTK), Eurocollins, and Celsior. However, there is no evidence that using a particular solution might reduce the incidence of DGF after KT [8]. The possibility to improve post-transplant outcomes using SCS with modified preservation solutions containing anti-thrombin perfluoro-carbon nanoparticles (PFC-NP) [18] or complement inhibitors such as APT070 [19] is under investigation. Among validated organ-targeted interventions, hypothermic machine perfusion (HMP) is certainly the most effective. Several randomized clinical trials and meta-analysis have demonstrated that HMP can reduce the incidence of PNF and DGF compared to SCS. Furthermore, superior one- and three-year transplant survival rates have been reported, particularly in ECD and DCD KT [20,21]. The proposed mechanisms of action are enhanced nitric oxide (NO) production, preserved adenosine triphosphate (ATP) synthesis, and increased resistance to oxidative stress, eventually determining reduced intra-vascular resistive index (RI), better intra-parenchymal perfusion, and less tissue damage [22]. Questioning routine organ preservation under hypoxic and hypothermic conditions, some authors have advocated the use of oxygenated preservation techniques [23] and sub-normo-thermic (SMP) or normo-thermic machine perfusion (NMP) [24]. Theoretical advantages of controlled oxygenation and heating are better mitochondrial viability and function, decreased production of reactive oxygen species (ROS), and reduced activation of cell death programs [24]. Whilst the usefulness of isolated organ oxygenation has not been confirmed in human studies [23], encouraging results have been obtained with oxygenation during HMP [25] and NMP [26]. In particular, current literature demonstrates that NMP is associated with significantly lower DGF rates than SCS and suggests that NMP can be safely used to assess and resuscitate poor quality kidneys from DCD donors [27].

Finally, recipient-related strategies are also available. Induction and peri-operative immunosuppressive regimens represent ideal targets of intervention. Lymphodepleting agents such as polyclonal rabbit anti-thymocyte globulin (rATG) or alemtuzumab (a monoclonal antibody targeting CD52) have been increasingly used in ECD and DCD KT to reduce the risk of rejection associated with IRI and minimize calcineurin inhibitor (CNI) exposure. Current data demonstrate excellent results in term of early acute rejection rates and successful CNI sparing, but the overall impact on DGF and long-term transplant survival remain marginal [2,3,28,29]. Peri-operative administration of complement inhibitors represent another feasible option as the complement system plays a key role in IRI, antigen presentation, and allograft rejection [6]. To date, clinical experience in KT is mostly limited to C5 inhibitors [30,31,32] and C1 esterase inhibitors (C1INH) [33,34,35]. Available studies in humans have shown that complement inhibition is effective in reducing post-transplant antibody-mediated rejection rates, but have no effects on the incidence and severity of DGF [32,35].

There is now mounting evidence that ASCs possess specific differentiative, regenerative, and immunomodulatory properties that could be used to prevent or repair IRI-associated organ damage and favor allograft acceptance. In particular, it has been demonstrated that ASCs are able to trans-differentiate into renal tubular epithelial cells (RTECs) [36]. Such ability is regulated by complex paracrine mechanisms [37]. Upon stimulation, ASCs can release multiple growth factors, chemokines, cytokines, and extra-cellular vesicles (EVs). Vascular endothelial growth factor (VEGF), basic fibroblast growth factor (bFGF), insulin-like growth factor 1 (IGF-1), platelet-derived growth factor (PDGF), hepatocyte growth factors (HGF), and transforming growth factor β1 (TGF-β1) have been recognized as the main mediators of the pro-angiogenic, immunomodulating, and anti-apoptotic effects of ASCs [38,39]. EVs also play a relevant role mediating inter-cellular communications and regulating the response to ischemia, inflammation, and antigen stimulation through the release of proteins, lipids, ssDNA, dsDNA, mRNA, siRNA, or miRNA [40,41].

ASCs-based therapies have already been used to treat autoimmune disorders including systemic sclerosis, Crohn’s disease, systemic lupus erythematosus, and graft-versus-host disease [42]. Encouraging results have also been observed in pre-clinical and clinical models of renal allograft IRI and rejection [43,44]. However, the relatively small number of studies available and their methodological limitations do not allow conclusions to be drawn regarding ASCs efficacy and safety. In fact, the exact mechanisms of action, optimal timing, doses, and routes of administration remain unclear.

In the present review, we summarize the current knowledge on the use of ASCs and ASC-derived EVs for the prevention and treatment of IRI and AR in KT. The possible utilization in organ reconditioning and repair is also highlighted.

## 2. Available Stem Cells Sources for Renal Tissue Repair

Various stem cells sources derived from embryonic or adult tissues have been proposed for the development of cell-based therapies in KT [45].

Embryonic stem cells (ESCs) are pluripotent stem cells with a high self-renewal capacity and with the possibility of multilineage differentiation [46]. It has been reported that ESCs can be induced to differentiate into RTECs, obtaining successful integration into engineered kidney-like organs and creating new renal tissue [47]. However, major legal and ethical issues as well as the risk of uncontrolled growth and teratoma formation limit the chances of future research and clinical application.

Induced pluripotent stem cells (iPSCs) represent an alternative source of pluripotent stem cells, sharing several characteristics with ESCs. Initially generated by reprogramming adult fibroblasts through the introduction of four transcription factors (OCT3/4, SOX2, c-MYC, and KLF4) [48], iPSCs have also been produced from keratinocytes, hepatocytes, dental pulp cells, mesangial cells, peripheral blood cells, cord blood, or extra-embryonic tissues [45]. In vitro studies have demonstrated that iPSCs can be used to produce bioengineered renal tissue [49] and kidney organoids [50]. In addition to legal and ethical issues, the main concerns about the use of iPSCs in clinical practice are the risk of aberrant differentiation with neoplasm formation and the need for viral vectors as gene transducers.

After many years of debate, recent studies have finally confirmed the existence of a renal stem/progenitor cells population in the adult kidney [51]. Proposed but unconfirmed cell markers for phenotypic characterization are CD24, CD133, CD106, vimentin, and PAX-2 [46]. Although the possibility of a like-for-like substitution appears as particularly intriguing, clinical use of these stem cells in KT presents several obstacles. Their undefined phenotype makes it extremely difficult to develop a standardized kidney isolation protocol. Moreover, the amount of renal tissue required for isolation and expansion might exceed the capacity of a single individual. Finally, it has been reported that renal stem/progenitor cells can undergo dysregulated proliferation and cause renal scarring, hyperplastic glomerular lesions, nephron loss, and degenerative diseases such as collapsing glomerulopathy or crescentic nephritis [52].

To date, mesenchymal stem/stromal cells (MSCs) represent the most extensively studied stem cells population in renal allograft IRI and tolerance induction models [53]. MSCs are multipotent cells with staminal features first isolated from the bone marrow in the 1960s and described as spindle-shaped, plastic-adherent elements with the ability to differentiate into adipocytes, osteocytes, or chondrocytes [54]. In the following years, similar cells have also been detected in the skeletal muscle, synovial membrane, dermis, placenta, cord blood, and peripheral blood. Remarkably, MSCs do not express HLA class II molecules and exhibit low-level expression of HLA class I [55] and costimulatory signal molecules such as CD80, CD86, CD40, and CD40L [56]. This specific immunological characteristic makes MCSs particularly suitable for allogenic cell-based therapies. Among MSCs, KT-related research has primarily focused on bone-marrow-derived stem/stromal cells (BM-MSCs) and ASCs [53]. BM-MSCs were discovered in 1996 and initially described as part of the stroma supporting hematopoietic stem cells [57]. In 2006, the International Society for Cell and Gene Therapy (ISCT) defined a standard cytofluorimetric panel to adequately profile MSCs [58]. According to this panel, BM-MSCs express CD105, CD73, CD90 and lack of CD45, CD34, CD14 or CD11b, CD79α or CD19, and HLA-DR [58]. BM-MSCs are usually harvested from the iliac crest with minimal discomfort for the patient. After successful evaluation in pre-clinical settings, BM-MSCs are now being investigated in numerous clinical trials [44].

In 2002, Zuk and colleagues found a particular population of MSCs located within the adipose tissue and named it ASCs [59]. ASCs reside in the peri-vascular niche, embedded in the extra-cellular matrix (ECM), next to macrophages, vascular endothelial cells, smooth muscle cells, fibroblasts, and pericytes, all being part of the stromal vascular fraction (SVF). The SVF can be easily obtained from lipoaspirates or adipose tissue biopsies through disruption of the ECM, using enzymatic digestion or mechanical separation. In vitro cultivation of the SVF allows to isolate and harvest ASCs in few weeks [60]. A joint statement of the International Federation for Adipose Therapeutics and Science (IFATS) and the ISCT has defined ASCs as phenotypically positive for CD13, CD29, CD44, CD73, CD90, and CD105 and negative for CD31 and CD45. ASCs can proliferate in vitro and differentiate into adipogenic, chondrogenic, and osteogenic lineages [61]. As a potential source of MSCs, the adipose tissue offers multiple advantages. The harvesting procedure is minimally invasive and less painful or risky than those required for BM-MSCs. Furthermore, it ensures a high cellular yield, also using the peri-renal fat tissue [62,63]. Remarkably, ASCs exhibit shorter doubling time, higher in vitro proliferation, longer life-span, and retarded senescence than BM-MSCs [64]. To date, the main concern regarding ASC-based therapies is long-term safety. Safety issues include genetic instability with possible neoplasms formation, differentiation toward undesired lineages, risk of micro-thrombosis in case of intra-arterial administration, risk of pulmonary, splenic or hepatic embolization when using intra-venous routes, and theoretical HLA sensitization after allogenic ASCs engraftment.

## 3. Mechanisms of Action

In order to explain the positive effects of ASCs on renal injury, two mechanisms of action have been proposed: differentiative and paracrine. In 2010, Li et al. demonstrated that, in a murine model of acute kidney injury (AKI), human ASCs (hASCs) injected in the tail vein 24 h after the induction of IRI could trans-differentiate into RTECs and repair the damage associated with ATN. At six months of follow-up, the animals treated with hASCs showed normal renal histology whilst the control group exhibited fibrosis and glomerular sclerosis [36]. ASCs’ ability to trans-differentiate into RTECs was confirmed by other studies in which mouse or rabbit kidney scaffolds were seeded with ASCs through the renal artery or ureter [65]. During these experiments, it was observed that ASCs were able to engraft the glomerular, tubular, and vascular areas of the scaffold, eventually trans-differentiating into RTECs and endothelial cells. The authors postulated that the attachment to the scaffold was mediated by the stromal cell-derived factor 1α (SDF-1α) [66]. It remains unclear whether ASCs could trans-differentiate into other renal cell types or build up a proper tridimensional structure. ASCs also exert complex paracrine actions [45]. The main soluble molecules produced in response to IRI and mediating ASCs’ pro-angiogenic, proliferative, and anti-apoptotic effects are IGF-1, SDF-1α, and VEGF [67,68]. HGF, IL-10, TGF-β1, and NO have been recognized as anti-inflammatory and anti-fibrosis agents [69] whereas tumor necrosis factor-inducible gene 6 protein (TSG-6) [70] and indoleamine 2,3-dioxygenase (IDO) [71,72] seem to promote the development of regulatory T-cells (Tregs) and induce tolerance via up-regulation of the FOX-P3 pathway [73].

EVs represent further mediators of ASCs’ function [74]. Classified as exosomes (30–100 nm) or micro-vesicles (100–1000 nm) depending on their maximal diameter, EVs can change their content and secretion pattern in response to different external stimulations [75,76]. Hypoxia, extra-cellular calcium variations, and thermal stress are powerful triggers of EVs secretion [77,78]. It has been hypothesized that ASC-induced renal tissue repair is mediated by specific mRNAs and miRNAs contained in the EVs [79]. In particular, miR-24, miR-29, and several members of the let-7 miRNA family might be involved in the regenerative, proliferative, anti-apoptotic, and anti-inflammatory response to IRI [80] whereas the immunomodulating effects on dendritic cells could be mediated by miR-21-5p, miR-142-3p, miR-223-3p, and miR-126-3p [81]. EV-related mRNAs can contribute to cell differentiation (RAX2, EPX, SCNN1G), immunomodulation (IL1RN, MT1X, CRLF1), ECM remodeling (COL4A2, IBSP), cell-cycle progress (SENP2, RBL1, CDC14B), angiogenesis, and fibrosis (TGF-β1, TGF-β3, FURIN, ENG) [82]. Investigating the role of EVs in AKI, Gao et al. observed that ASC-derived exosomes could significantly decrease the expression of important inflammatory mediators, such as NF-κB, p65 protein, and TNF-α [83]. Exosomes can also up-regulate SIRT-1, thus protecting RTECs from oxidative stress [84]. There is evidence that ASCs’ EVs are involved in cell survival regulation and tissue repair. As a matter of fact, exosomes release is associated with increased expression of the pro-survival protein BCL2 and decreased expression of the pro-apoptotic caspase-3, caspase-9, and BAX [83]. Exosomes also favor angiogenesis induction through up-regulation of HIF-1α and VEGF [83]. According to a recent meta-analysis of pre-clinical rodent models of AKI, ASC-derived EVs administration improves renal function, reducing apoptosis and inflammation [85]. Remarkably, some reports suggest that EV-induced renal protection from IRI exhibits a concentration-dependent effect [41].

Main ASCs and ASC-derived EVs mechanism of actions in IRI and KT models are summarized in Figure 1 and Figure 2.

## 4. Harvesting Procedures

ASCs reside in peri-vascular niches within the adipose tissue, constituting about 20% of the cellular population of the SVF. Two main ASCs harvesting techniques are currently available: lipoaspirate and adipose tissue biopsy. Following specimen collection, the ECM of the adipose tissue undergoes enzymatic or mechanical disruption. The SVF released during the procedure is seeded and expanded in vitro to obtain ASCs [59]. Cultivation and expansion allow to produce a large number of ASCs from a relatively small sample of adipose tissue and permit to collect purified cellular products [61]. However, these methods are expensive, time-consuming, and require specific Good Manufacturing Practice (GMP) facilities. Therefore, it is unlikely that they can be used in the peri-operative setting of a deceased donor KT [43]. Allogenic non-donor derived ASCs represent an intriguing option as off-the-shelf cellular products. Nonetheless, some authors remain concerned that prolonged culture may cause cellular senescence and favor the acquisition of a pro-inflammatory phenotype [86].

The process required for SVF extraction is easy and expeditious enough that it can be performed in a timely manner in most hospitals. The SVF is a composite cellular mixture with wide intra- and inter-individual variability in cells quality and quantity, depending on patient’ characteristics, harvesting procedure, extraction technique, and isolation method [87]. Such heterogeneity represents a remarkable issue as it negatively affects standardization and reproducibility of the vast majority of research protocols involving the SVF. Furthermore, the amount of adipose tissue required to obtain an adequate number of ASCs is significantly greater using the SVF extraction method than the ASCs expansion technique. Lastly, it is worth considering that the effects exerted on the transplanted kidney by the other cell types retrieved during the procedure (namely, pericytes, preadipocytes, endothelial cells, and fibroblasts) are still unclear [43]. Immunomodulatory, anti-inflammatory, and pro-tolerogenic properties have been postulated for the non-staminal cells of the SVF such as adipose resident M2 macrophages, Tregs, and regulatory invariant natural killer (NK) cells, but they have not been confirmed. [88,89,90].

The use of stem cell-based therapies in solid organ transplantation raises the question of whether autologous or allogenic cells should be preferred. Due to the lack of HLA class II molecules and the low expression of HLA class I and co-stimulatory signal molecules, ASCs are considered as immune-privileged cells [91]. However, the actual risk of allograft rejection associated with the use of allogenic ASCs remains obscure [92,93,94]. It has been demonstrated that donor-derived ASCs can promote the development of Tregs in vitro [73]. A certain degree of tolerance induction has also been also observed in vivo [95]. Nevertheless, a study comparing the efficacy of autologous or allogenic ASCs in a rat model of KT with total MHC mismatch, showed that donor-derived ASCs administration did not improve allograft survival and was associated with increased mortality through an immunologically-mediated mechanism [96]. Considering the paucity of data, it sounds reasonable to assume that autologous ASCs represent a safer option compared to allogenic ASCs. Some authors argue that, due to the toxic effects of uremia, ASCs harvested from ESRD patients have lower regenerative potential than those retrieved from healthy subjects; thus limiting the use of autologous ASC-based therapies in KT setting. However, there is evidence that the beneficial properties of ASCs are not affected by uremia or dialysis. In fact, Roemeling-van Rhijn and colleagues demonstrated that ASCs harvested from patients with CKD exhibited normal characteristics and function. Remarkably, ASCs’ proliferative capacity and genetic stability remained consistent over time, also after repeated expansion and prolonged exposure to uremic serum. Furthermore, no differences could be detected between patients with normal or impaired renal function in ASCs’ response to immune activation and ASCs’ inhibitory effect on the proliferation of alloantigen-activated peripheral blood mononuclear cells [97].

In theory, the use of ASC-derived EVs could help address many issues associated with traditional cell-based therapies such as the need for time-consuming and technically demanding preparation processes and the risk of HLA sensitization. A recent meta-analysis of EV-based therapies in pre-clinical rodent models of AKI concluded that MSC-derived EVs administration was able to improve renal function, modulate the inflammatory response [98], and reduce cell apoptosis [85]. Overall, available studies suggest that EVs could be beneficial for the prevention and treatment of renal IRI [99,100]. On the contrary, current literature does not seem to support the utilization of EV-based therapies for KT rejection [96]. The highly variable composition of different EVs mixtures should also be considered as a potential limitation [77].

## 5. Timing, Dose, and Route of Administration

The proper definition of the optimal timing, dose, and route of administration represents a key factor for the implementation of ASC-based therapies in KT. Possible routes of administration that can maximize ASCs homing and survival in the allograft include intra-venous, intra-arterial, and intra-parenchymal. Continuous infusion using a kidney perfusion machine has also been proposed. There is evidence that MSCs and ASCs homing to the wounded areas of the kidney is promoted by a specific chemoattractant. Among the pathways involved in MSCs migration during IRI, the SDF-1/CXCR4 axis plays a major role. Over-expression of the CXCR4 gene enhances MSCs migration and promotes neo-angiogenesis and vascular repair [101,102,103]. These signals also regulate ASCs homing from the injection site to the transplanted organ [104].

The easiest route of administration of ASCs is intra-venous injection. However, when injected into the venous circulation, ASCs undergo a first pass effect in “filter organs” such as the liver, spleen, or lung. This phenomenon reduces the number of cells reaching the target site and can cause engulfment of the pulmonary micro-circulation, eventually leading to pulmonary embolism and death [105]. The results of clinical trials evaluating efficacy and safety of intravenously administered ASCs for the treatment of graft-versus-host disease demonstrate that the risk of micro-thrombosis and micro-embolism depends on the total number of cells infused. According to these studies, doses ranging from 0.4 to 9 × 10^6^ MSCs or ASCs per kilogram represent a safe option [105,106,107].

Intra-arterial administration can also be considered. In theory, ASCs injection into the renal artery could increase the number of cells engrafting the kidney. A meta-analysis of MSC-based therapies in small animal models of AKI concluded that the intra-arterial route was associated with better renal function preservation (as assessed by serum creatinine concentration, SCr) than the intra-venous route [108]. It seems that, following injection into the renal artery, ASCs remain on site through passive mechanisms, mostly disappearing within 14 days [109]. Sporadic episodes of mesangioproliferative glomerulonephritis with aberrant adipocytic differentiation have been reported [110]. It can also be argued that, compared to intra-venous injection, intra-arterial administration is more invasive, entailing a greater risk of complications. Nevertheless, it should be recognized that a KT procedure offers wide kidney exposure and easy access to the renal artery.

ASCs intra-parenchymal injection represents another feasible option and it proved effective in several animal models of AKI [67]. This route of administration allows to combine ASCs with special biomaterials so as to extend their permanence and survival in the target site. An elegant study by Gao et al. showed that the use of thermosensitive chitosan chloride hydrogel (CCH) could improve the efficacy of ASC-based therapy in IRI. More in details, CCH was able to increase intra-renal ASCs retention and survival as well as to promote host kidney cells proliferation and resistance to apoptosis. At four weeks, the groups treated with CCH exhibited higher renal micro-vessels density and RTECs proliferation rates than controls, with better renal function. A possible explanation is that CCH favors a more suitable micro-environment for ASCs survival through enhanced ROS scavenging [111]. Encouraging results were also obtained by Zhou and colleagues using renal ECM hydrogel as an injectable scaffold to deliver ASCs into ischemic kidneys. In their experience, ECM hydrogel reduced ASCs dispersion and increased the number of viable ASCs retained in the target site. Furthermore, the authors observed that ECM hydrogel could inhibit ASCs apoptosis in vitro and promote ASCs proliferation, trans-differentiation into RTECs, and secretion [112]. Overall, available data suggest that ASCs cultivation in three-dimensional aggregates or spheroids combined with direct injection into the renal cortex can increase ASCs retention in the kidney and improve the efficacy of ASC-based therapies. Compared to bi-dimensional monolayers, three-dimensional configurations provide better resistance to oxidative stress and ischemia. They also facilitate cell migration and secretion [113,114].

ASCs can be delivered to the renal allograft using specific perfusion machines [115]. Several randomized clinical trials and meta-analysis have demonstrated that HMP reduces the incidence of DGF compared to SCS, particularly in case of ECD or DCD transplants [116]. Therefore, combining the benefits of continuous or pulsatile kidney perfusion with the regenerative potential of ASCs-based therapy represents a promising strategy. Preliminary data obtained from animal models of KT or acute kidney IRI showed that adding ASCs, BM-MSCs, or BM-MSC-derived EVs to the machine perfusion solution could improve renal function and allograft survival [117,118,119]. Current experience with NMP is more limited. Indeed, higher costs and logistic difficulties prevent widespread utilization of this preservation modality. Pool and coworkers investigated the effects of NMP plus hASCs in a model of porcine DCD KT. The experimental treatment was administered following kidney preservation with SCS [120] or HMP [68]. Interestingly, positive results were observed only in the HMP group. Compared to SCS, HMP ensured better ASCs homing and viability as well as higher resistance to oxidative stress and improved tubular regeneration capacity.

Other crucial aspects of ASC-based therapies not yet clarified are timing and frequency of administration. According to a meta-analysis of small animal models of renal IRI, a single dose of ASCs administered after ischemia provides the best outcomes [108]. On the contrary, studies on tolerance induction and rejection suggest that repeated ASCs administrations after transplant are more effective than a single dose of ASCs [96,121]. Probably, in the setting of allogenic transplantation, the huge antigenic mass provided with prolonged treatments favors chimerism and allograft acceptance whereas a small antigenic load, as the one provided with a single dose of donor-derived cells, acts as an additional antigen-specific stimulation. However, experience with BM-MSCs does not seem to support this hypothesis as the vast majority of studies on tolerance induction [122,123,124] and IRI [124,125] showed that pre-transplant administration could offer the best window of opportunity. Indeed, BM-MSCs infusion before transplant was associated with enhanced cells migration toward secondary lymphoid organs, reduced intra-parenchymal inflammation, and increased MSC-induced Tregs expansion whilst post-transplant infusion failed to achieve tolerance induction and renal function preservation, determining a proinflammatory micro-environment. [124,125].

Possible ASC-based therapies routes, timing, and frequency of administration in IRI and KT models are summarized in Figure 3 and Figure 4.

## 6. Pre-Clinical Studies

### 6.1. Ischemia–Reperfusion Injury

One of the first studies investigating ASCs administration in AKI was performed by Li et al. in 2010. In their seminal work, the authors evaluated the efficacy of intravenously injected hASCs in a mouse model of renal IRI. They observed that hASCs were able to promptly differentiate into RTECs, reduce inflammation, and completely repair the damaged areas of the kidney [36]. Using a similar model, another research group found that most intravenously injected ASCs reached the lung rather than the kidney, thus revealing an endocrine mechanism of action. As a matter of fact, although distributed in different organs, ASCs could exert a positive effect on ATN and renal inflammation, as assessed by mononuclear cells infiltration and cytokine/chemokine concentrations (IL-1b, IL-17, MIP-1α, RANTES, and IP-10) [104]. An interesting study performed in a male Sprague-Dawley rat model of IRI assessed sequential ASCs administration using different infusion techniques. A first dose of ASCs was injected into the renal artery during the ischemic phase of the experiment whereas two more doses of ASCs were intravenously administered at six and 24 h of observation. Compared to controls, the rats treated with ASCs showed lower SCr and blood urea nitrogen (BUN) concentrations, less renal tissue damage, higher vascular cells density, and increased angiogenesis. ASCs infusion was associated with reduced expression of several biomarkers of inflammation, apoptosis, and oxidative stress such as NAD(P)H quinone oxidoreductase 1 (NQO1), heme-oxygenase (HO-1), glutathione peroxidase (GPx), and glutathione reductase (GR) [126]. Positive results were also obtained using a Wistar rat model of renal IRI. Rats were randomized to receive placebo, a single shot of ASCs administered through the tail vein, or two sequential doses of ASCs delivered via intra-parenchymal injection. Renal biopsies collected at 31 days of follow-up showed improved tubular vascularization and reduced degree of ATN in both experimental arms of the study. However, intra-parenchymal administration was associated with better histology (tubular injury score) and improved ASCs engraftment [127]. Another study compared intra-venous and intra-arterial injection of a single dose of ASCs in a rat model of IRI. ASCs were administered at the time of reperfusion. The two infusion techniques proved equivalent in terms of renal function, inflammation, apoptosis, and oxidative stress. Remarkably, in both groups, ASCs were detected not only in renal micro-circulation but also in pulmonary vessels. The beneficial effects of ASCs were found to be incremental up to a total dose of 5 × 10^5^ cells. Using 1 × 10^7^ cells, all benefits were lost, probably due to the formation of ASCs aggregates or plugs in renal vessels. Following intra-venous infusion of 5 × 10^7^ cells, all animals died of massive pulmonary vessels occlusion [105]. Experience with ASC-based therapies in kidney allograft IRI models is more limited. In the study performed by Iwai and colleagues in 2014, Lewis rats were transplanted using isogenic DCD kidneys preserved for one hour in extracellular-trehalose-Kyoto (ETK) solution under sub-normothermic conditions. A single shot of 1 x 10^6^ ASCs was administered through the allograft artery prior implantation or via intra-venous injection immediately after transplant. On post-operative day four, both experimental groups showed lower SCr and BUN concentrations than untreated rats. However, only intra-arterial administration was associated with a survival benefit (75% vs. 50%) [117]. Combined treatments using ASCs and cyclosporine were also explored, with conflicting results [103,128].

Given the fact that obtaining expanded ASCs can be logistically challenging in transplant setting, some authors advocate the use of freshly isolated SVF. Although limited, available data suggest that SVF intra-arterial injection is able to reduce the damage associated with acute kidney IRI, favoring tissue repair and function [129]. Direct comparison between ASCs and SVF in a Sprague-Dawley rat model of xenotransplantation supports the use of SVF. In the study performed by Zhou et al., hASCs or SVF were administered via intra-parenchymal injection. After 72 h, both treatments groups exhibited lower SCr and BUN concentration than placebo. ASCs and SVF effects on apoptosis, tubular injury, inflammation, and neo-angiogenesis were also equivalent [67].

It has been postulated that the efficacy of ASCs-based therapies could be improved preconditioning the kidney with hypoxia or ischemia. Kang and colleagues demonstrated that hypoxia was able to increase ASCs survival and paracrine secretion ex vivo [130]. In a rat model, renal preconditioning through a combination of ischemia (three cycles of two minutes) and reperfusion (five minutes) followed by ASCs administration 60 min before the induction of IRI, significantly increased the efficacy of ASCs. Indeed, lower BUN concentrations and higher intra-renal levels of several marker of cell proliferation and angiogenesis (HIF-α1, SDF-1α, CD31, and Ki67) were observed in preconditioned rats treated with ASCs compared to those receiving ASCs without preconditioning [131]. Few reports suggest that ASCs can be preconditioned ex-vivo and successfully used at a later stage in vivo, with enhanced survival and function [132]. Interestingly, it seems that ASCs preconditioning with hypoxia can increase the production and secretion of EVs [130]. Moreover, there are data demonstrating that EVs collected after ASCs preconditioning can exert more powerful anti-apoptotic, pro-angiogenic, anti-oxidative, and immunomodulatory effects [77]. It is plausible that ASCs and ASC-derived EVs may act synergistically. To test this hypothesis, Lin and colleagues assessed the efficacy of a combination of ASCs and ASC-derived EVs in a rat model of IRI. As controls, they used rats treated with ASCs or ASC-derived EVs. Within 72 h of administration, the group receiving the combination therapy showed better renal function, improved kidney injury score, and higher expression of anti-apoptotic, pro-angiogenic, and anti-oxidative biomarkers than single-treatment groups. Accordingly, biomarkers related to oxidative stress, inflammation, and fibrosis were significantly lower [133].

Most relevant studies on ASCs-based therapies in KT IRI models are summarized in Table 1.

### 6.2. Tolerance Induction and Rejection

It has been recognized that IRI-induced inflammation leads to increased HLA expression, greater antigen-presenting cells (APCs) effectiveness, and higher donor-specific T cells reactivity [6].

Collett and colleagues investigated the effects of hASCs administration on renal histology in a rat model of IRI, particularly focusing on inflammation and T cells infiltration. They found that hASCs injection into the supra-renal aorta could significantly increase rats’ survival and mitigate capillary rarefaction in the cortex and outer medulla of the kidney. It was also observed that hASCs were able to exert immunomodulating effects and profoundly alter the quality and quantity of infiltrating T cells. Precisely, hASCs administration was associated with reduced T CD4+ IL-17+ or T CD8+ IL-17+ pro-inflammatory lymphocytes and increased T CD25high/FOXP3+ regulatory lymphocytes [135]. Tregs play a major role in promoting graft tolerance [138]. Therefore, the ability of ASCs to favor the expansion of the Tregs pool in the recipient deserves further research [69]. To date, how ASCs might increase the total number of donor-specific Tregs remains unclear. In particular, it is still debated whether such increase is due to clonal expansion of natural Tregs or phenotype-switch of effector T cells. Mixing donor-derived ASCs and recipient-derived CD25 -/dim effector T cells in vitro, Engela et al. demonstrated that donor-derived ASCs could induce the formation of functional de novo Tregs, increasing the proportion of CD25+/CD127- FOXP3+ cells in the subset of T CD4+ lymphocytes within the pool of CD25-/dim allostimulated T cells. Remarkably, donor-derived ASC-induced de novo Tregs were able to inhibit effector T cells proliferation just like natural Tregs. The methylation of the Treg-specific demethylated region (TSDR) in the FOXP3 gene supports the hypothesis that de novo Tregs were induced rather than natural. Moreover, ASC-mediated Tregs induction was inhibited by IL-2 blockade, suggesting an IL-2-dependent mechanism [73]. In another study evaluating the interaction between ASCs and Tregs, it was observed that ASCs and Tregs could coexist and exert a synergistic action on effector T cells. On the contrary, donor-derived and recipient-derived Tregs could not affect the function of donor-derived ASCs. Indeed, Tregs were not able to inhibit the production of interferon-γ (IFN-γ), which is fundamental for ASCs activation and immunomodulation. The expression and release of IDO, another major mediator of ASCs-induced effector T cells inhibition, were also not influenced by Tregs, much like the ability of ASCs of reducing the secretion of TNF-α or promoting the production of IL-10 [69]. These results in vitro were confirmed by several studies in vivo. Kato et al. tested the infusion of autologous ASCs in a rat model of KT with total MHC mismatch. Dark Agouti rat kidneys were transplanted into nephrectomized Lewis rats, injecting 2 × 10^6^ recipient-derived ASCs into the renal artery of the donor before organ retrieval. Autologous ASCs were able to inhibit alloreactive T cells proliferation and prolong allograft survival. Lower levels of mRNAs encoding for IL-2, TNF-α, and IFN-γ as well as increased production of TSG-6 were detected in animals treated with ASCs. Among these molecules, TSG-6 was recognized as the most plausible mediator of ASCs-induced immunomodulation. Preliminary data showed that TSG-6 could suppress the proliferation of alloreactive T cells through down-regulation of CD44 on T CD4+ cells surface; thus reducing their ability to trigger rejection [70].

Multiple schemes of MSCs or MSC-derived EVs administration were evaluated in a model of Fisher-to-Lewis rat KT. The study compared infusions of autologous BM-MSCs or ASCs, autologous BM-MSC-derived or ASC-derived EVs, donor BM-MSCs or ASCs, and donor BM-MSC-derived or ASC-derived EVs. Donor cells or EVs were administered as a single shot immediately after transplant to reduce post-transplant recipient sensitization. Autologous treatments were administered according to a multiple-dose regimen (at the time of transplant, 4 weeks, and 8 weeks after transplant). Administration of autologous or donor EVs did not have any impact on allograft function and survival. Donor cell-based treatments also failed to demonstrate any beneficial effect. Rather, the injection of donor ASCs was associated with increased mortality. Only rats treated with autologous BM-MSCs or ASCs exhibited improved allograft function. Eventually, all transplants failed due to rejection. However, histological evaluation demonstrated that the groups receiving autologous cells had lower degrees of tubulitis, capillaritis, and interstitial fibrosis. A reduction in the number of infiltrating T and B lymphocytes as well as NK cells was also described [96].

In the study by Liu et al., ASCs were transfected with a recombinant plasmid inducing the expression of the fusion protein OX40-Ig [139]. OX40 is a secondary co-stimulatory immune checkpoint molecule predominantly expressed on activated T CD4+ cells and inhibition of the OX40/OX40 ligand signal pathway has successfully been used for the treatment of autoimmune diseases. In a rat model of allogenic KT, administration of OX40-Ig-expressing ASCs was associated with increased infiltrating Tregs and reduced infiltrating effector T cells than treatment with non-modified ASCs. Furthermore, rats receiving OX40-Ig-expressing ASCs showed lower rejection rates and improved allograft survival. Reduced intra-graft IFN-γ expression levels and up-regulation of several immunomodulation-related mRNAs, such as IL-10, TGF-β1, and FOXP3 were observed in the experimental arm [139].

A synthetic overview of the most relevant pre-clinical studies evaluating ASC-based therapies for tolerance induction or prevention of allograft rejection is presented in Table 2.

## 7. Clinical Studies

Current literature on the use of MSCs as tolerance induction agents in clinical transplantation is extremely scarce [44] and actually refers to the experience of a single research group [95,140]. In their first report, these authors described the outcomes of a prospective open-label comparative clinical trial in which 285 living donor KT recipients were allocated to a tolerance induction regimen based on the combination of autologous hematopoietic stem cells (HSCs) plus autologous ASCs or autologous HSCs only. Both regimens were administered 14 days before transplant through the portal vein. Patients receiving HSCs plus ASCs showed significantly higher recipient and death-censored graft survival rates than those treated with HSCs only. Furthermore, both experimental treatments were associated with lower acute rejection rates and reduced maintenance immunosuppression than controls [140]. In another prospective trial with no control group and a very small population of 10 living donor KT recipients, patients were treated with a combination of autologous HSCs plus autologous ASCs injected into the portal vein 14 days before transplant and an immunosuppressive induction regimen including rATG, rituximab, bortezomib, and steroids. After six years of follow-up, 50% of the subjects enrolled were free from maintenance immunosuppression with excellent allograft function, thus demonstrating achievement of full tolerance toward the transplant kidney [95].

## 8. Conclusions

The prevention and treatment of renal IRI and allograft rejection represent major challenges for the transplant community. Over the years, several strategies have been proposed with rather disappointing results. More recently, there has been a renewed interest in exploring cell-based therapies. Among the options available, ASCs may be the ideal choice as they are relatively easy to obtain and potentially able to trans-differentiate into RTECs. Relevant anti-oxidative, anti-inflammatory, anti-apoptotic, pro-angiogenic, and tolerogenic properties have also been demonstrated both in vitro and in vivo. The current literature supports the hypothesis that ASCs activity is regulated by complex paracrine or perhaps endocrine mechanisms. As a matter of fact, it has been observed that, upon appropriate stimulation, ASCs can release growth factors, chemokine, cytokines, NO, and IDO. Mounting evidence suggests that the production and secretion of EVs play a pivotal role in regulating ASCs inter-cellular cross-talk as well as their response to ischemia, oxidative stress, and inflammation.

Clinical experience with ASCs or ASC-derived EVs in KT is anecdotal and focused on tolerance induction. Although interesting, these studies have substantial flaws and lack of reproducibility. The present knowledge on the use of ASC-based therapies in IRI, AKI, and allograft rejection mostly derives from animal models. Pre-clinical studies have shown that ASCs administration can mitigate the impact of renal IRI, improving kidney cells resistance to oxidative stress, reducing inflammation, and favoring tissue repair or regeneration. Positive results have been reported exploring ASC-based tolerance induction strategies. Indeed, it has been demonstrated that allogenic ASCs can promote the formation of antigen-specific Tregs and inhibit the proliferation of antigen-specific effector T cells in the recipient. Important safety information has also been obtained. In particular, several reports have shown that intra-venous administration and total doses exceeding 1 × 10^7^ cells can cause severe vascular complications, such as renal micro-thrombosis, pulmonary embolism, and death. The theoretical risks of malignant transformation and allosensitization should not be neglected.

To date, ASCs in KT setting remain a promising but unconfirmed therapeutic option. There are several critical issues to be addressed before the encouraging results observed in pre-clinical models can translate into clinical practice. Future research projects should primarily focus on the optimal route and dose of administration.

## Figures and Tables

**Figure 1 ijms-22-11188-f001:**
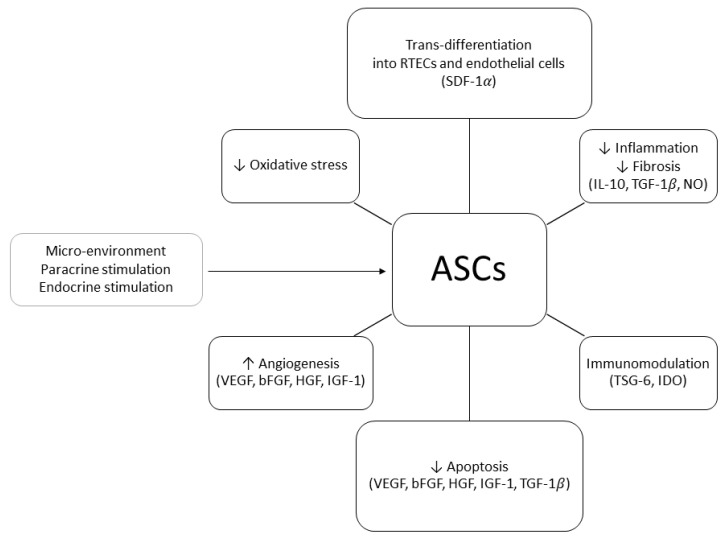
Adipose-derived stem/stromal cells (ASCs) mechanisms of action in kidney transplant ischemia–reperfusion injury and tolerance induction models (↑, increased; ↓, decreased; RTEC, renal tubular epithelial cell; SDF-1, stromal cell-derived factor 1; IL-10, interleukin 10; TGF-1, transforming growth factor 1; NO, nitric oxide; TSG-6, tumor necrosis factor-inducible gene 6; IDO, indoleamine 2,3-dioxygenase; VEGF, vascular endothelial growth factor; bFGF, basic fibroblast growth factor; HGF, hepatocyte growth factor; IGF-1, insulin-like growth factor).

**Figure 2 ijms-22-11188-f002:**
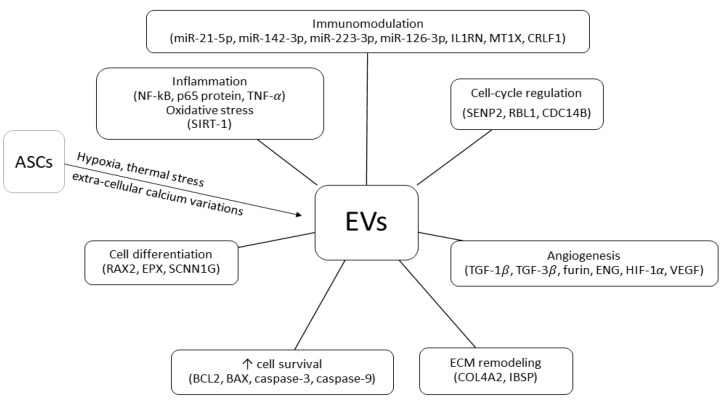
Adipose-derived stem/stromal cell extra-cellular vesicles (EVs) mechanisms of action in kidney transplant ischemia–reperfusion injury and tolerance-induction models (↑, increased; NF-kB, nuclear factor kappa-light-chain-enhancer of activated B cells; p65 protein, REL-associated protein; TNF, tumor necrosis factor; SIRT-1, sirtuin-1; RAX2, retina and anterior neural fold homeobox protein 2; EPX, eosinophil peroxidase; SCNN1G, sodium channel epithelial 1 subunit gamma; BCL2, B-cell lymphoma 2; BAX, bcl2-like protein 4; COL4A2, collagen type IV alpha 2 chain; IBSP, integrin binding sialoprotein; TGF-1, transforming growth factor 1; TGF-3, transforming growth factor 3; ENG, endoglin; HIF-1, hypoxia-inducible factor-1; VEGF, vascular endothelial growth factor; SENP2, SUMO specific peptidase 2; RBL1, retinoblastoma-like protein 1; CDC14B, dual specificity protein phosphatase CDC14B; miR, micro RNA; IL1RN, interleukin 1 receptor antagonist; MT1X, metallothionein 1X; CRLF1, cytokine receptor like factor 1).

**Figure 3 ijms-22-11188-f003:**
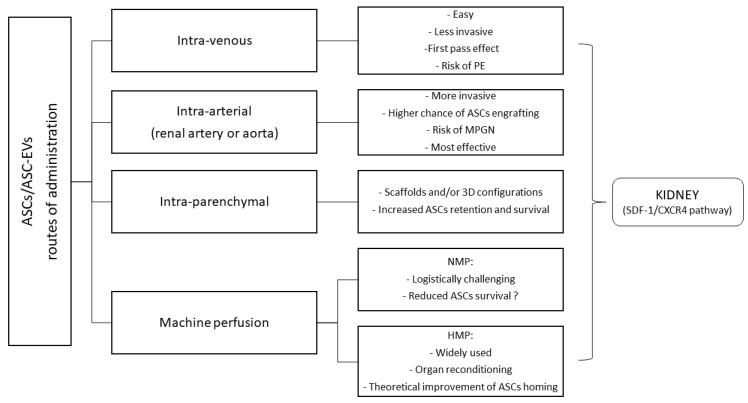
Adipose-derived stem/stromal cells (ASCs) and ASCs-derived extra-cellular vesicles (EVs) routes of administration in kidney transplant ischemia–reperfusion injury and tolerance-induction models (PE, pulmonary embolism; MPGN, membranoproliferative glomerulonephritis; NMP, normo-thermic machine perfusion; HMP, hypo-thermic machine perfusion; SDF-1, stromal cell-derived factor 1; CXCR4, C-X-C motif chemokine receptor 4).

**Figure 4 ijms-22-11188-f004:**
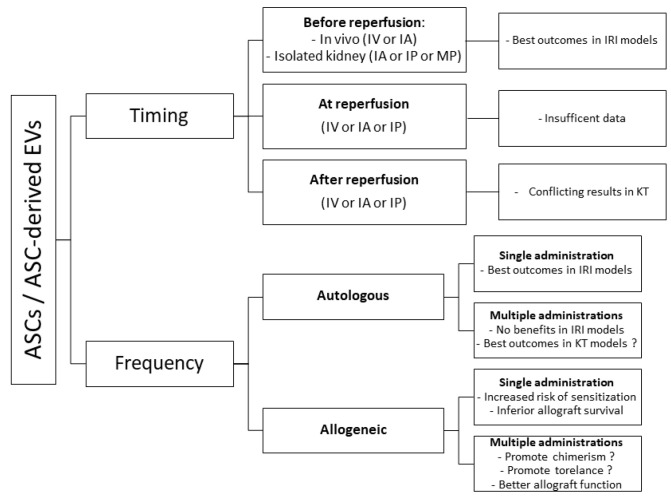
Adipose-derived stem/stromal cell (ASC)-based therapies timing and frequency of administration in kidney transplant (KT) ischemia–reperfusion injury (IRI) and tolerance-induction models (EVs, extra-cellular vesicles; IV, intra-venous; IA, intra-arterial; IP, intra-parenchymal; MP, machine perfusion).

**Table 1 ijms-22-11188-t001:** Pre-clinical studies evaluating the use of adipose-derived stem cells in renal ischemia–reperfusion injury models.

Authors Year Reference	Model Experimental Model	Source	Control	Dose	Route	Timing	Main Findings
Li et al. 2010 [36]	C57BL/6 mice In vivo IRI (35 min ischemia)	mBM-MSC vs. hASCs	Yes	5 × 10^5^ cells	IV	Within 24 h of reperfusion	- hASCs favor renal tissue repair - hASCs and mBM-MSCs have similar regenerative potential - hASCs can differentiate into RTECs - hASCs reduce tissue inflammation
Chen et al. 2011 [126]	Male Sprague-Dawley rats In vivo IRI (1h ischemia)	Aut. rASCs	Yes	1 × 10^6^ cells	IP+IV	IP: at reperfusion + IV: 6 h, 24 h post-reperfusion	- Aut. rASCs improve renal function and histology - Aut. rASCs reduce inflammation, apoptosis, and oxidative stress - Aut. rASCs increase angiogenesis
Furuichi et al. 2012 [104]	C57BL/6 mice In vivo IRI (45 min ischemia)	mASCs	Yes	1 × 10^5^ cells	IV	At reperfusion 24 h, 48 h post-reperfusion	- mASCs improve renal function - mASCs reduce tubular injury and macrophage infiltration - Following IV injection, most mASCs remain trapped in the lung
Gao et al. 2012 [111]	Male Sprague-Dawley rats In vivo IRI (40 min ischemia)	rASCs in PBS vs. rASC in CCH	Yes	2 × 10^6^ cells	IP	10 min post-reperfusion	- CCH improves rASCs retention and survival in the target site - CCH+rASCs adm. increases host renal cells proliferation - CCH+rASCs adm. reduces host renal cells apoptosis - CCH+rASCs adm. improves renal function recovery
Chen et al. 2013 [103]	Male Sprague-Dawley rats In vivo IRI (Ischemia time N/A)	rASCs+CsA vs. rASC vs. CsA	Yes	1.2 × 10^6^ cells	IV	1 h, 6 h, 24 h post-reperfusion	- rASCs+CsA adm. is more effective than rASCs - rASCs+CsA adm. reduces inflammation, apoptosis, and oxidative stress - rASCs+CsA adm. increases neo-angiogenesis and ROS scavengers - rASCs+CsA adm. improves renal function recovery
Shih et al. 2013 [105]	Male Sprague-Dawley rats In vivo IRI (45 min ischemia)	rASCs	Yes	5 × 10^4^ cells vs. 1 × 10^5^ cells vs 5 × 10^5^ cells vs. 1 × 10^7^ cells vs. 5 × 10^7^ cells	IV vs. IA	At reperfusion	- IV and IA injection have equivalent effects on rASCs function - IV and IA rASCs adm. improve renal function recovery - IV and IA rASCs adm. reduce inflammation, apoptosis, oxidative stress - rASCs positive effects are incremental up to a total dose of 5 × 10^5^ cells - IV rASCs total doses exeeding 5 × 10^7^ cells can be lethal
Wang et al. 2013 [118]	Wistar inbred rats In vivo cold IRI (30 min ischemia + cold solution)	Aut. rASCs	Yes	2 × 10^6^ cells	IP+IV	IP: at reperfusion + IV: 6 h post-reperfusion	- Aut. rASC improve renal function and histology - Aut. rASCs reduce inflammation and apoptosis
Iwai et al. 2014 [117]	Male Lewis rats Marginal isogenic DCD KT (SNT preservation + ETK solution)	rASCs	Yes	1 × 10^6^ cells	IV vs. IA+ETK	IV: at KT (Rec.) vs. IA: 3 h prior KT (graft)	- rASCs IV adm. does not improve Rec. survival - rASCs IA adm. improves Rec. survival
Zhang et al. 2014 [132]	Male Sprague-Dawley rats In vitro + In vivo IRI (40 min ischemia)	Hypo-rASCs vs. rASCs	Yes	2 × 10^6^ cells	IP	At reperfusion	- Hypo-rASCs are more effective than rASCs - Hypo-rASCs improve renal function and histology - Hypo-rASCs reduce apoptosis and oxidative stress - Hypo-rASCs increase neo-angiogenesis - Hypoxia reconditioning enhances rASCs paracrine secretion in vitro - Hypoxia reconditioning increase rASCs resistance to oxidative stress
Hussein et al. 2016 [131]	Male Sprague-Dawley rats In vivo IRI (45 min ischemia)	Aut. Isch-rASCs vs. Aut. rASCs	Yes	1 × 10^6^ cells	IV	60 min prior ischemia	- Aut. Isch-rASCs are more effective than Aut. rASCs - Aut. Isch-rASCs and Aut. rASCs improve renal function - Aut. Isch-rASCs increase renal cells proliferation and neo-angiogenesis - Ischemia preconditioning improve Aut. rASCs efficacy
Lin et al. 2016 [133]	Male Sprague-Dawley rats In vivo IRI (1h ischemia)	Allo ASCs + ASC-EVs vs. Allo ASCs vs. ASC-EVs	Yes	ASCs: 1.2 × 10^6^ cells EVs: +100 μg	IV	3 h post-reperfusion	- Allo ASCs + ASC-EVs is more effective than Allo ASCs or ASC-EVs - Allo ASCs + ASC-EVs adm. improves renal function and histology - Synergistic effect of ASCs and ASC-EVs on tissue repair
Rosselli et al. 2016 [134]	Shorthair cats In vivo IRI (1 h ischemia)	Allo cASCs vs. Allo cBM-MSCs	Yes	4 × 10^6^ cells	IV	1 h post-reperfusion	- All treatments failed to show any beneficial effect on renal function
Zhao et al. 2016 [113]	Male Sprague-Dawley rats In vitro + In vivo IRI (50 min ischemia)	2D rASCs vs. 3D rASCs aggregates	Yes	1 × 10^6^ cells	IP	10 min post-reperfusion	- 3D aggregates improve rASCs engraftment and function - 3D rASCs more effective than 2D ASCs in preserving renal function
Xu et al. 2016 [114]	Male Sprague-Dawley rats In vitro + In vivo IRI (40 min ischemia)	2D hASCs vs. 3D hASCs spheroids	Yes	2 × 10^6^ cells	IP	At reperfusion	- 3D spheroids improve hASCs function - 3D hASCs more effective than 2D hASCs in preserving renal function - 3D hASCs adm. is associated with better renal histology than 2D hASCs
Collett et al. 2017 [135]	Male Sprague-Dawley rats In vivo IRI (40 min ischemia)	hASCS	Yes	2 × 10^6^ cells	IA	At reperfusion	- hASCs increase rat survival and promote renal function recovery - hASCs increase Tregs infiltration in the kidney -hASCs reduce Th17 cells infiltration in the kidney
Lee et al. 2017 [129]	Male Sprague-Dawley rats In vivo IRI (40 min ischemia)	Aut. SVF	Yes	N.A.	IP vs. IA	At reperfusion	- SVF IA adm. promotes renal function recovery - SVF IA adm. more effective than SVF IP adm.
Zhou et al. 2017 [67]	Male Sprague-Dawley rats In vivo IRI (45 min ischemia)	hSVF vs. hASCs	Yes	hSVF: N/A hASCs: 2 × 10^6^ cells	IP	At reperfusion	- hSVF and hASCs are equally effective in preserving renal function - hSVF and hASCs exert similar effects on renal histology - hSVF reduces inflammation, apoptosis, and tubular injury - hSVF promotes neo-angiogenesis
Zhang et al. 2017 [136]	Male Sprague-Dawley rats In vivo IRI (40 min ischemia)	rASCs	Yes	2 × 10^6^ cells	IV	4 h post-reperfusion	- rASCs completely repair the damaged areas of the kidney - rASCs exhibit anti-apoptotic and anti-inflammatory properties
Monteiro et al. 2018 [127]	Female Wistar rats In vivo IRI-model (24 h ischemia)	rASCs	Yes	1.2 × 10^6^ cells	IV vs. IP	24 h post-reperfusion vs. 48 h post-reperfusion	- IP rASCs adm. more effective than IV rASCs adm. - Late (48 h) rASCs adm. more effective than early (24 h) rASCs adm. - Late (48 h) IP ASCs adm. provides optimal renal function preservation
Collino et al. 2019 [77]	Male Wistar rats In vitro + In vivo IRI (45 min of ischemia)	Hypo-hASC-EVs vs. hASC-EVs	Yes	7.5 × 10^8^ EVs	IP	At reperfusion	- Hypoxia preconditiong enhances hASCs secretory activity - Hypoxia preconditioning affects hASC-EVs secretion pattern - Hypo-hASC-EVs and hASC-EVs improve renal function recovery - Hypo-hASC-EVs exhibit higher pro-angiogenetic potential
Hafazeh et al. 2019 [137]	Male Wistar rats In vivo IRI (45 min ischemia)	Aut. rASCs	Yes	2 × 10^6^ cells	IV	At reperfusion	- Aut. rASCs improve renal function recovery
Pool et al. 2019 [120]	Swine Ex vivo study Isolated porcine kidney Cold storage + NMP + hASCs	hASCs	Yes	1 × 10^5^ cells vs. 1 × 10^6^ cells vs. 1 × 10^7^ cells	NMP	1 h after NMP	- NMP does not improve hASCs engraftment and survival - Prolonged NMP is associated with increased hASCs death - No functional information provided
Sierra et al. 2019 [109]	Danish Landrace-X-Yorkshire pigs In vivo IRI (1h ischemia)	sASCs	Yes	1 × 10^7^ cells	IA	At reperfusion	- IA adm. favors sASCs retention through passive mechanisms - After IA, sASCs remain on site for 14 days
Pool et al. 2020 [68]	Swine Ex vivo study Isolated porcine kidney (20 min ischemia) HMP (2–3 h) + NMP (7 h)	hASCs vs. hBM-MSCs	Yes	hASCs: 1 × 10^7^ cells hBM-MSCs: 1 × 10^7^ cells	NMP	1 h after NMP start	- hASCs and hBM-MSCs decrease renal injury markers levels - hASCs and hBM-MSCs increase perfusate immunomoduling cytokines
Zhou et al. 2020 [112]	Male Sprague-Dawley rats In vitro + In vivo IRI (45 min ischemia)	rASCs + renal ECMH vs. rASCs	Yes	2 × 10^6^ cells	IP	Prior IRI induction	- ECMH improves rASCs retention and survival in the kidney - ECMH reduces rASCs diffusion to other organs and tissues - ECMH enhances rASCs regenerative capacity

Abbreviations: IRI, ischemia–reperfusion injury; min, minute; mBM-MSC, murine bone-marrow mesenchymal stem cell; hASC, human adipose-derived stem cells; IV, intra-venous; RTEC, renal tubular epithelial cell; h, hour; Aut., autologous; rASC, rat adipose-derived stem cell; IP, intra-parenchymal; mASC; murine adipose-derived stem cell; PBS, posphate-buffering saline; CCH, chitosan chloride hydrogel; CsA, cyclosporine; adm., administration; DCD, donation after circulatory death; KT, kidney transplant; SNT, sub-normo-thermic; EKT, extra-cellular-trealose-Kyoto; adm.; administration; Rec., recipient; Hypo-rASCs, hypoxia-preconditioned rat adipose-derived stem cell; Aut., autologous; Isch-rASCs, ischemia-preconditioned rat adipose-derived stem cell; Allo, allogeneic; ASC-EV, ASC-derived extra-cellular vesicle; cASC, cat adipose-derived stem cell; cBM-MSC, cat bone marrow mesenchymal stem cell; 2D, two-dimensional; 3D, three-dimensional; Tregs, regulatory T cell; Th17, T helper 17; SVF, stromal vascular fraction; Hypo-hASC-EV, hypoxia-preconditioned human adipose-derived stem cell extra-cellular vesicle; hASC-EV, human adipose-derived stem cell extra-cellular vesicle; X, cross-bred; sASC, swine adipose-derived stem cell; hBM-MSC, human bone-marrow mesenchymal stem cell; ECMH, extra-cellular matrix hydrogel; NMP, normo-thermic machine perfusion.

**Table 2 ijms-22-11188-t002:** Pre-clinical studies evaluating the use of adipose-derived stem/stromal cells for tolerance induction or rejection prophylaxis in kidney transplantation.

Authors Year Reference	Model	Source	Control	Dose	Route	Timing	Main Findings
Engela et al. 2013 [73]	In vitro MLR and SA	Don. ASCs + Rec. CD25-/dim eff. T cells	No	N/A	N/A	N/A	- Don. ASCs induce de novo Tregs formation - ASC-induced de novo Tregs originate from CD25+/CD127- FOXP3+ cells - IL-2 dependent mechanism
Engela et al. 2013 [69]	In vitro MLR and SA	Don. ASCs + Rec. Tregs	No	N/A	N/A	N/A	- Don. ASCs and Rec. Tregs act synergistically - Don. ASCs favor Rec. Tregs development and function - IL-10 dependent mechanism
Kato et al. 2014 [70]	In vitro MLR + Rat KT	Aut. ASCs vs. Placebo	Yes	2 × 10^6^ cells	IA	Aut. ASCs adm. to the Don. before kidney retrieval	- Aut. ASCs inhibit antigen-specific T cells proliferation in vitro and in vivo - Aut. ASCs adm. to the Don. before kidney retrieval prolong graft survival - Aut. ASCs suppress alloreactive T cells proliferation - TSG-6 dependent mechanism (CD44 down-regulation)
Liu et al. 2017 [139]	In vitro MLR + Rat KT	Aut. ASCs expressing OX40-Ig vs. Aut. non-modified ASCs	Yes	2 × 10^6^ cells	IV + IP	IV: 4d before KT + IP: at KT + IV: 6 h post-KT	- Aut. ASCs (modified and non-modified) increase Tregs - Aut. ASCs (modified and non-modified) reduce effector T cells - Aut. ASCs expressing OX40-Ig improve graft survival and reduce rejection
Ramirez et al. 2020 [96]	Rat KT	Aut. ASCs vs. Aut. BM-MSCs Aut. ASC-EVs vs. Aut. BM-MSC-EVs Don. ASCs vs. Don. BM-MSCs Don. ASC-EVs vs. Don. BM-MSC-EVs	Yes	1 × 10^6^ cells 1.4 × 10^9^ EVs	IV	Don. cells or EVs: IV shot at KT Aut. cells or EVs: IV shot at KT, 4w, 8w post-KT	- ASC-EVs and BM-MSC-EVs (Don. and Aut.) do not affect graft outcomes - Don. ASCs and BM-MSCs do not affect graft outcomes - Don. ASCs are associated with increased mortality - Aut. ASCs and BM-MSCs improve early graft function - Aut. ASCs and BM-MSCs reduce infiltrating T, B, and NK cells

Abbreviations: MLR, mixed lymphocyte reaction; SA, suppression assay; Don., donor; ASC, adipose-derived stem cell; Rec., recipient; N/A, not available; Treg, regulatory T cell; Aut., autologous; KT, kidney transplant; IA, intra-arterial; adm., administration; IV, intra-venous; IP, intra-parenchymal; BM-MSC, bone marrow mesenchymal stem cell; EV; extra-cellular vesicle.

## Data Availability

No new data were created or analyzed in this study. Data sharing is not applicable to this article.

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
