# Peer review of "Adipose-Derived Stem/Stromal Cells in Kidney Transplantation: Status Quo and Future Perspectives"

_ijms, 2021, doi:10.3390/ijms222011188_

Round 1

Reviewer 1 Report

Dear All,

In the review: “Adipose-derived stem/stromal cells in kidney transplantation: status quo and future perspectives”, Storti et al. summarized the current knowledge on the use of adipose stem/stromal cells and adipose stem/stromal cells-derived extracellular vesicles for the prevention and treatment of ischemia-reperfusion injury and acute rejection after kidney transplantation.

Please, take into consideration the following major remarks:

  1. The authors should mention the strategies used nowadays to mitigate the impact of IRI and favor tolerance, even if they have disappointing results. This is necessary to highlight the importance of the proposed technique. Maybe it will be useful to add some details related to cost-effectiveness of ASCs treatment, compared to the current techniques, and very important the comfort of the patients (e.g. it is less/more painful and so on).
  2. There is lack of appropriate representation through figures. Because it is a review, I think you should have at least 3-4 original images, so try to organize some data as figures/ diagrams. In this way the manuscript will be easier to read and understand.
  3. Being a complex review, I find it useful to add an abbreviations section.
  4. Manuscript needs thorough proofreading to correct the typographical and grammatical errors.

Author Response

We are grateful to Reviewer 1 for his/her positive feedback and valuable support.

The manuscript has been revised according to the comments.

More in details:

Q1a) The authors should mention the strategies used nowadays to mitigate the impact of IRI and favor tolerance, even if they have disappointing results. This is necessary to highlight the importance of the proposed technique. 

A1a) As suggested, we added an entire paragraph in the introduction section of the manuscript describing main strategies for the prevention of IRI-induced organ damage and transplant rejection (lines 57-109, highlighted in yellow).

Q1b) Maybe it will be useful to add some details related to cost-effectiveness of ASCs treatment, compared to the current techniques, and very important the comfort of the patients (e.g. it is less/more painful and so on).

A1b) We agree with Reviewer 1 that it would be interesting to explore cost-effectiveness of currently available strategies and ASC-based treatments. However, there is a generalized lack of data on this specific topic and direct comparison between different techniques is almost impossible. Brief information regarding procedure-related risks and patient satisfaction have been added in the paragraphs describing different stem cells populations and administration routes.

Q2) There is lack of appropriate representation through figures. Because it is a review, I think you should have at least 3-4 original images, so try to organize some data as figures/ diagrams. In this way the manuscript will be easier to read and understand.

A2) As suggested, we included 4 Figures in the revised version of the manuscript.

Q3) Being a complex review, I find it useful to add an abbreviations section.

A4) As suggested, we added an abbreviations section at the end of the revised version of the manuscript and in all figure legends.

Q4) Manuscript needs thorough proofreading to correct the typographical and grammatical errors.

A4) We extensively checked the manuscript for typo and grammatical errors. 

Reviewer 2 Report

I read with interest the manuscript entitled “Adipose-derived stem/stromal cells in kidney transplantation: status quo and future perspectives” by Gabriele, et al that is intended to be published in IJMS as a review.

The review is excellent and elegant. Authors address the various positive values and limitations of using ASC.

The main concern is that a naïve reader may believe that there only exist ASC and not other type of stromal cells. Or that only ASC are effective in renal transplantation. I agree that the review is focused on ASC but it needs a chapter where authors discuss differences between the several kinds of stem cells.

There are some studies, for instance using BM-MSC which give valuable information that may be shared with ASC. For instance, concerning timing of infusion, as reported by several authors as https://doi.org/10.3727/096368917x695010 from Merino A, et al, or 10.1111/tri.12132, from Perico N, et al, which should be referenced (in lines 94 95, 187 188, 285 286 or 307 308)

In this sense, there are some information concerning ASC which should be shared:

  • Mesenchymal stromal cells and kidney transplantation: pretransplant infusion protects from graft dysfunction while fostering immunoregulation. Some information points to inflammation exacerbation depending on the moment of infusion, specially post transplantation
  • Furthermore, ASCs exhibit shorter doubling time, higher in vitro proliferation, longer life-span, and retarded senescence. These characteristics favors ASC but do not distinguish quantitatively or qualitatively to other cells
  • ASCs may be the ideal choice as they are relatively easy to obtain. This is not absolutely certain, bone marrow is a huge source of MSC, especially from the donors.

Author Response

We are grateful to Reviewer 2 for his/her positive feedback and valuable support.

As requested, we revised the manuscript addressing all the comments.

More in details,

Q1) The main concern is that a naïve reader may believe that there only exist ASC and not other type of stromal cells. Or that only ASC are effective in renal transplantation. I agree that the review is focused on ASC but it needs a chapter where authors discuss differences between the several kinds of stem cells.

A1) As suggested, we added an entire chapter describing other sources of stromal cells potentially useful for the development of cell-based therapies in kidney transplantation (chapter 2, lines 169-238, highlighted in green).

Q2) There are some studies, for instance using BM-MSC which give valuable information that may be shared with ASC. For instance, concerning timing of infusion, as reported by several authors as https://doi.org/10.3727/096368917x695010 from Merino A, et al, or 10.1111/tri.12132, from Perico N, et al, which should be referenced (in lines 94 95, 187 188, 285 286 or 307 308)

A2) As requested, we added the aforementioned references in the revised version of the manuscript. A brief discussion of shared information was also included (lines 497-504, highlighted in green).

Q3) In this sense, there are some information concerning ASC which should be shared: "Mesenchymal stromal cells and kidney transplantation: pretransplant infusion protects from graft dysfunction while fostering immunoregulation. Some information points to inflammation exacerbation depending on the moment of infusion, specially post transplantation".

A4) As suggested, we further discussed the topic in the revised version of the manuscript (lines 486-504).

Q5) Furthermore, ASCs exhibit shorter doubling time, higher in vitro proliferation, longer life-span, and retarded senescence. These characteristics favors ASC but do not distinguish quantitatively or qualitatively to other cells. ASCs may be the ideal choice as they are relatively easy to obtain. This is not absolutely certain, bone marrow is a huge source of MSC, especially from the donors.

A6) In order to better describe currently available MSCs populations with pros and cons, we added an entire new chapter in the revised version of the manuscript (Chapter 2, lines169-238, highlighted in green).

Round 2

Reviewer 1 Report

The authors adequately addressed the comments in the revised manuscript.

Reviewer 2 Report

A huge job. The review is comprehensive but very understandable. It is an excellent contribution to this theme.